# Data-Driven Lipschitz Continuity: A Cost-Effective Approach to Improve Adversarial Robustness

## Abstract

The security and robustness of deep neural networks (DNNs) have become increasingly concerning. This paper aims to provide both a theoretical foundation and a practical solution to ensure the reliability of DNNs. We explore the concept of Lipschitz continuity to certify the robustness of DNNs against adversarial attacks, which aim to mislead the network with adding imperceptible perturbations into inputs. We propose a novel algorithm that remaps the input domain into a constrained range, reducing the Lipschitz constant and potentially enhancing robustness. Unlike existing adversarially trained models, where robustness is enhanced by introducing additional examples from other datasets or generative models, our method is almost cost-free as it can be integrated with existing models without requiring re-training. Experimental results demonstrate the generalizability of our method, as it can be combined with various models and achieve enhancements in robustness. Furthermore, our method achieves the best robust accuracy for CIFAR10, CIFAR100, and ImageNet datasets on the RobustBench leaderboard.

## 1 Introduction

Deep neural networks (DNNs) have demonstrated promising results across various tasks (Krizhevsky et al., 2012; Redmon et al., 2016), prompting concerns about AI security as these networks are increasingly deployed in our daily lives. A single erroneous prediction could lead to catastrophic consequences. For example, the Overload attack can significantly inflate the inference time of detecting objects for self-driving systems (Chen et al., 2023), while even minor typos in input prompts can cause large language models to produce unexpected responses (Zhu et al., 2023).

The focus of this paper is to design robust DNNs that can defend against adversarial attacks, which aim to create perturbations in inputs that are imperceptible to humans but can mislead DNNs. Previous studies have revealed the existence of adversarial examples in diverse domains, such as image pixels (Szegedy et al., 2013), audio data (Carlini & Wagner, 2018), and textual content (Li et al., 2018). Consequently, exploring the vulnerabilities of DNNs and developing theoretically grounded explainable AI is crucial for ensuring the reliability of DNN-based applications.

Adversarial training (Madry et al., 2017) has proven to be an effective strategy for enhancing the robustness of DNNs. It achieves this by generating adversarial examples on the fly during the training phase and optimizing the model's weights to minimize the losses caused by these examples. Recent studies have shown that robustness can be further improved by introducing additional examples from other datasets (Carmon et al., 2019) or using generative models (Gowal et al., 2021; Wang et al., 2023) to cover low-frequency data. Despite the promising improvements in robustness, training costs increase significantly due to the demand for additional data, which can be up to 20 to 100 times larger than the original dataset. This poses a trade-off between training cost and robustness. The concern over high computational costs becomes a significant obstacle in deploying robust DNN-based applications, especially in fields like medicine, autonomous driving systems, and other areas where human lives are at stake.

In this paper, we explore how robustness is certified by the theorem of Lipschitz continuity, which theoretically gauges how much outputs are amplified by the perturbations. However, we argue that

$$\begin{pmatrix} x_1 \\ x_2 \\ x_3 \\ x_4 \\ x_5 \\ x_6 \end{pmatrix} \xrightarrow{f^{\text{forge}}} \begin{pmatrix} x_1 \\ x_2 \\ x_3 \\ 0 \\ 0 \\ 0 \end{pmatrix} \rightarrow \boxed{\begin{array}{l} \text{Linear systems} \\[4pt] \text{Conv}(\,\cdot\,) \\ \text{FC}(\,\cdot\,) \end{array}}$$

$$f^{\text{forge}} = \begin{cases} 0, & \|x\| \leq c \\ x, & \text{otherwise} \end{cases}$$

Figure 1: The empirical Lipschitz constant of specific layers that can be represented by linear systems, such as convolutional or fully connected layers, can be reduced by remapping their input domain to a constrained range.

the set of observed data is finite and cannot cover the entire real data space, leading to an overestimation of the Lipschitz constant derived from the theorem. Therefore, we propose an algorithm that can remap the input domain into a constrained range, resulting in a Lipschitz constant for the modified function that is less than or equal to the Lipschitz constant of the original function, thus potentially enhancing robustness. Our key contributions are outlined as follows:

- We introduce the concept of the empirical Lipschitz constant, which can more precisely reflect the robustness of the corresponding observed data. Compared with the original definition of Lipschitz constant, the empirical value is derived from a set of observed data, thereby eliminating the influence of space that is never drawn from real data. As illustrated in Figure 1, we prove that any function that can be formulated as a linear system, when combined with our proposed function to remap the input domain of a specific layer to a constrained range, can reduce its empirical Lipschitz constant, resulting in better robustness.

- Our method is well-suited for inference-time model optimization to enhance robustness cost-effectively. The proposed function can enhance the robustness of adversarially trained models with minimal additional costs. Specifically, it introduces only one parameter, the value of which can be determined by scanning an observed data once without the need for re-training or fine-tuning.

- The experimental results suggest that our method can be combined with various existing methods and gain robustness improvements. Besides, our method achieves the best robust accuracy against adversarial examples generated by AutoAttack (Croce & Hein, 2020), a state-of-the-art ensemble attack, for CIFAR10, CIFAR100, and ImageNet datasets on the RobustBench leaderboard (Croce et al., 2020). By assessing accuracy against adaptive attacks, transfer attacks, and evaluation methods for validating obfuscated gradients (Athalye et al., 2018; Carlini et al., 2019), we believe that the proposed algorithm should not cause robustness to be overestimated.

The rest of this paper is organized as follows. Section 2 introduces the background on adversarial attacks and adversarial training. Section 3 presents the theoretical proof of how robustness is enhanced by manipulating the domain of linear functions and introduces the proposed algorithm. Section 4 shows the experimental resultsand ablation studies on various hyper-parameters, combination with different activation functions and gradient masking verification. The last section is our conclusion.

## 2 RELATED WORKS

### 2.1 ADVERSARIAL ATTACKS

Adversarial attacks aim to inject tiny perturbations into inputs, causing victim DNNs to output incorrect predictions with high confidence (Chen & Hsieh, 2022). These attacks have been observed in numerous vision applications (Goodfellow et al., 2014; Chen et al., 2018; Wang et al., 2022; Yin

et al., 2022). Furthermore, these tiny perturbations can be embedded not only in image pixels but also in textual contexts (Kumar et al., 2023; Yao et al., 2023), audio space (Xie et al., 2021), and other fields (Ilahi et al., 2021). Some research has shown how adversarial attacks threaten real applications (Xu et al., 2020; Komkov & Petiushko, 2021; Du et al., 2022; Wei et al., 2022). Investigating the vulnerability of DNNs and theoretically avoiding adversarial examples when optimizing model weights or designing architecture is an ongoing challenge.

Adversarial attacks can be classified into two types based on the amount of information the attacker has access to: white-box attacks and black-box attacks. In the case of white-box attacks, the attacker has full access to all information about the victim model. Methods such as the PGD attack (Madry et al., 2017) and AutoAttack (Croce & Hein, 2020) generate adversarial examples by leveraging the gradient direction of the model. Although this scenario is often unrealistic in practical settings, research in this area is valuable for developing more robust models in the future.

In contrast, black-box attacks, such as the square attack (Andriushchenko et al., 2020) and ZO-NGD (Zhao et al., 2020), only provide the attacker with access to the model's output predictions. In these cases, adversarial examples can still be generated through transferability, where models with similar architectures are used to create adversarial examples (Wang et al., 2021; Chen et al., 2024). The goal of black-box attacks is to investigate the potential risks of adversarial attacks in real-world application scenarios, where attackers may have limited access to model internals.

## 2.2 DEFENSIVE STRATEGIES

Adversarial training is a defensive strategy that aims to find optimal weights against adversarial attacks. It achieves this by generating adversarial examples on the fly during the training phase and optimizing the model's weights to minimize the losses caused by these examples. Despite the superior robustness achieved by adversarial training, the associated training costs of adversarially trained models are generally ten times more expensive than those of models trained utilizing a standard policy. The concern over high computational costs becomes a significant obstacle in deploying DNN-based applications.

Balancing between training cost and robustness is a challenge for adversarial training. Fast adversarial training has been proposed for applications pursuing higher robustness under a limited budget (Chen & Lee, 2020; Zhang et al., 2022). However, numerous adversarial examples cannot be drawn from these approaches, potentially leading to catastrophic overfitting, where robust accuracy significantly decreases without warning signs (Rice et al., 2020). On the contrary, some studies attempted to refine robustness by introducing additional examples from other datasets (Carmon et al., 2019) or using generative models (Gowal et al., 2021; Wang et al., 2023). Alternatively, another line of research has demonstrated that the removal of partial adversarial examples does not compromise robust accuracy, addressing the issue of unaffordable training costs (Zhang et al., 2020; Chen & Lee, 2024).

Despite the potential of adversarial training to enhance model robustness, budgetary constraints often limit the scope of their crafting to one or two specific attack types during the training stage. This restricted approach may inadvertently render adversarially trained models susceptible to novel, unseen attacks. As an alternative, Lipschitz-based certified training offers a theoretical framework for ensuring an upper bound on prediction errors (Gowal et al., 2018; Huang et al., 2021; Müller et al., 2022). However, it is important to acknowledge that these training methods often suffer from scalability issues.

## 3 METHODOLOGY

### 3.1 MOTIVATION

In this paper, we approach robustness from a theoretical perspective, aiming to demonstrate that all risks posed by adversarial examples are limited while minimizing the additional costs associated with improving robustness. Our evaluation is conducted under the white-box scenario, where the target model is capable of defending against various types of known adversarial attacks, including white-box attacks (Madry et al., 2017; Chen et al., 2018; Croce & Hein, 2020), black-box attacks (Chen et al., 2017), and transfer attacks (Demontis et al., 2019; Qin et al., 2022). Additionally, we

conduct a set of experiments to verify that gradient masking (Athalye et al., 2018) does not occur in our method and to ensure that robustness is not overestimated.

## 3.2 LIPSCHITZ CONTINUITY

To achieve our goal, we introduce a quantitative metric known as the Lipschitz constant, which gauges how much outputs are amplified by the perturbations within the input domain. The mathematical definition is as follows, a function $f : \mathbb{R}^m \to \mathbb{R}^n$ is globally Lipschitz continuous if there exists an constant $K \geq 0$ such that

$$D_f(f(x_1), f(x_2)) \leq K D_x(x_1, x_2) \quad \forall x_1, x_2 \in \mathbb{R}^m, \tag{1}$$

where $D_x$ is a metric on the domain of $f$; $D_f$ is a metric on the range of $f$; and $x_1 \neq x_2$. For a DNN, it can be considered as a composite function:

$$F(x) = (f_1 \circ f_2 \circ \cdots \circ f_L)(x), \tag{2}$$

where $f_i$ is the function of $i$-th layer. If there exists a Lipschitz constant for each individual layer, we can derive an upper bound of the Lipschitz constant for the victim model as follows,

$$K_F \leq \prod_{i=1}^{L} K_i, \tag{3}$$

where $K_i$ is the Lipschitz constant of $f_i$.

By defining adversarial examples $x^{\text{adv}}$ within a $\epsilon$-ball centered at an image $x$ as the inputs of (1), we can assess the impact caused by adversarial examples. Therefore, the Lipschitz constant serves as a bridge that connects the design of robust models with the measurement of risks posed by adversarial examples. A small Lipschitz constant for the victim model implies that the increase in loss is minimal, indicating a higher ability to resist adversarial attacks. Consequently, the objective of this paper is to lower the upper bound of Lipschitz constant for the given models.

As indicated by previous studies (Yoshida & Miyato, 2017; Farnia et al., 2018), Lipschitz constant of the given model defined in (3) can be minimized by reducing the output discrepancy of individual linear layers. Under the $L_2$ norm, we have

$$\frac{||f(x^{\text{adv}}) - f(x)||_2}{||x^{\text{adv}} - x||_2} = \frac{(||Wx^{\text{adv}} + b) - (Wx + b)||_2}{||\delta||_2} = \frac{||W\delta||_2}{||\delta||_2}, \tag{4}$$

where $W$ is the weight matrix; and $\delta$ is the distance between $x^{\text{adv}}$ and $x$. Therefore, the original optimization problem of minimizing Lipschitz constant is transformed into the following minimization problem:

$$\min_{W} \max_{\delta \neq 0, \delta \in \mathbb{R}^m} \frac{||W\delta||_2}{||\delta||_2} = \min_{W} \sigma_{\max}(W), \tag{5}$$

where $\sigma_{\max}(W)$ represents the largest singular value of the matrix $W$. Notably, there is a relation to eigenvalues:

$$\sigma_i^2(W) = \lambda_i(WW^\dagger) = \lambda_i(W^\dagger W), \tag{6}$$

where $W^\dagger$ is the conjugate transpose of $W$. Each singular value of the matrix $W$ is the square root of the eigenvalue of the matrices $WW^\dagger$ or $W^\dagger W$. In other words, minimizing $\lambda_{\max}(WW^\dagger)$, the largest eigenvalue of the matrices, can achieve the same objective.

Rather that minimizing the objective directly, Gershgorin circle theorem provides an alternative solution to estimate the robustness of the given linear system.

**Theorem 1.** *(Gershgorin Circle Theorem) For an $m \times m$ matrix $A$ with entries $a_{ij}$, each eigenvalue of $A$ is in at least one of the disk:*

$$R_i = \{z \in \mathbb{C} : |z - a_{ii}| \leq \sum_{i \neq j} |a_{ij}|\} \quad \text{for} \quad i = \{1, 2, \ldots, m\}. \tag{7}$$

Theorem 1 indicates each row vector can be represented as a disk which is centered at the diagonal entry $a_{ii}$ and whose radius is the sum of the off-diagonal entries $a_{ij}$. For any layer which can be represented by a linear system, such as convolutional or fully connected layers, robustness can be improved by shrinking the radius of the disk with the largest eigenvalue.

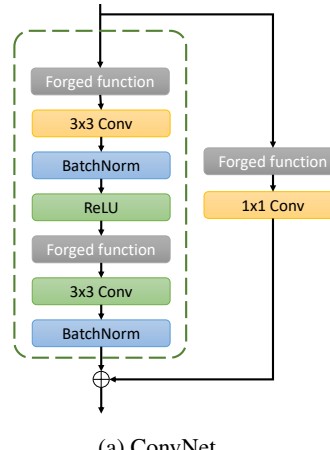
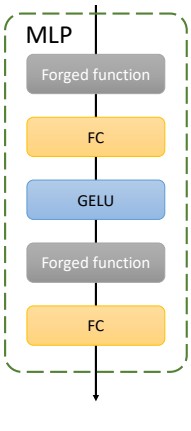

(a) ConvNet (b) Transformer

Figure 2: Insertion points of the forged function. In ConvNets, it is inserted into the residual blocks, while in Transformers, it is inserted into the MLP layers.

### 3.3 FORGED FUNCTION

We argue that the largest singular value provides a loose bound for the Lipschitz constant. To precisely reflect the robustness of the corresponding observed data, we define the empirical Lipschitz constant that eliminates the influence of space that is never drawn from real data.

**Definition 1.** *Empirical Lipschitz constant:*

$$\max_{\delta \neq 0, x \in \mathcal{S}} \frac{||Wx||_2}{||x||_2} \quad \forall x \in \mathcal{S}, \tag{8}$$

where $\mathcal{S}$ is an observed dataset. As can be seen, the empirical Lipschitz constant on the finite dataset is less than or equal to its Lipschitz constant derived from the theorem.

Based on Definition 1, we can build robust models by manipulating the output ranges of individual layers, thereby restricting the input domain of the next layer. If input vectors do not align with the direction of the eigenvector with the largest eigenvalue, the empirical constant should be bounded. Therefore, we proposed a forged function defined as follows:

$$f^{\text{forge}}(x) = \begin{cases} 0 & \text{if} \quad |x| \leq c_i^{\text{th}}, \\ x & \text{otherwise}, \end{cases} \tag{9}$$

where $c_i^{\text{th}}$ is a threshold for the $i$-th layer. Compared with the original functions, the range of the forged function is suppressed if its value is less than the threshold. When $c_i^{\text{th}}$ is set to 0, the forged function degrades into the original function.

The forged function aim to reduce the empirical Lipschitz constant of the layers that can be represented as linear systems by remapping the input domain of these layers into a constrained set. Figure 2 provides a visual representation of potential insertion points for the forged function, while maintaining the integrity of other layers. For the ResNet architecture, the forged function is placed before the convolutional layers in each residual block. Similarly, for vision transformer architectures, the structure of MLP layers is adapted to seamlessly integrate the forged function.

Here is the proof that the largest eigenvalue can be shrunk by the forged function. Let $W$ be the weight of the target layer, which can be represented by an $m \times n$ matrix, and $\mathbf{t}$ be the input vector. Without loss of generality, we assume that $A = W^\dagger W$ and $f^{\text{forge}}(\mathbf{t})$ is defined as:

$$f^{\text{forge}}(t_i) = \begin{cases} 0 & i \leq k \\ t_i & \text{otherwise}, \end{cases} \tag{10}$$

where $t_i$ is the $i$-th element of $\mathbf{t}$ and $k$ is a positive number.

**Lemma 2.** *There exists a matrix $A'$ whose largest eigenvalue, $\lambda_{max}(A')$, is less than or equal to the largest eigenvalue of $A$, $\lambda_{max}(A)$, if*

$$A f^{forge}(\boldsymbol{t}) = A'\boldsymbol{t}. \tag{11}$$

*Proof.* Since the first $k$ entries of the vector $\mathbf{t}$ are replaced with zeros, above condition can be achieved by replacing the corresponding column vectors of the matrix $A$ with zero vectors. Therefore, the entries of $A'$ are formulated as

$$a'_{ij} = \begin{cases} 0 & j \leq k \\ a_{ij} & \text{otherwise.} \end{cases} \tag{12}$$

The matrix $A$ is a positive semidefinite matrix, implying that the diagonal entries are non-negative. Moreover, with the entry representation of $A'$ in (12), we observe that modifications are only applied to the first $k$ columns, while the rest remain unchanged. Combining the Gershgorin Circle Theorem, we know that the centers of the first $k$ disks of the matrix $A'$ are shifted towards zero. Additionally, the radii of all disks, the absolute values of the off-diagonal entries in $A'$, are shrunk. Consequently, the upper bound of the largest eigenvalue of the matrix $A'$ is tighter compared to that of the original matrix $A$. □

Notably, the outputs of $f^{\text{forge}}(\mathbf{t})$ vary depending on the inputs, resulting in each input having its own $A'$. The upper bound of the largest eigenvalue of each matrix $A'$ must be not greater than the largest eigenvalue of the matrix $A$. With Lamma 2, a precise upper bound of the largest eigenvalue can be obtained by feeding a set of observed images. On the contrary, there might be cases in which solving the minimization problem in (5) leads to the theoretical largest eigenvalue being minimized, but the empirical Lipschitz constant remains unchanged.

The choice of a proper $c_i^{\text{th}}$ is a crucial factor in reducing the largest eigenvalue. In this paper, we propose obtaining the value of $c_i^{\text{th}}$ through the following equation:

$$c_i^{\text{th}} = c^r \max(F_{1 \to i}(x)) \quad \forall x \in \mathcal{S}, \tag{13}$$

where $\mathcal{S}$ can include all or a subset of images in the training set, $c^r$ is a positive number and $F_{1 \to i}(x)$ represents the output of the $i$-th layer. Specifically, each layer has its own $c_i^{\text{th}}$, but they share the same hyper-parameter $c^r$. Algorithm 1 specifies the implementation details of the forged function. The variable $b$ is used to store the maximum value that appeared in $\mathcal{S}$, as defined in (13), and is initialized during construction. Similar to the implementation of the batchnorm layer, the behavior is depended on the mode configuration. When the mode is set to tracking mode, the variable $b$ is updated accordingly, and the input is set to the output without any modification. Conversely, when the mode is set to inference mode, the value of $b$ is frozen, but the input is updated as defined by (9). By default, the mode is set to inference, and the values of $b$ and $c^r$ are zero, respectively. As a result, the set $\mathcal{M}$ is empty, and the algorithm is degraded to the identical function.

It is worth emphasizing that by feeding all images in the set $\mathcal{S}$ once in track mode beforehand, the value of $c_i^{\text{th}}$ can be obtained. The elements satisfying the constraints are appropriately deactivated during inference. Notably, this operation does not necessitate gradient computations and incurs minimal time consumption, typically only a few minutes, even when executed on commonly used GPUs. In comparison to adversarial training, this process is nearly cost-free. The overall procedure shares many similarities with post-pruning techniques. Nevertheless, we posit that the proposed function is very similar to the ReLU function, as it suppresses the output values within a specific range, but the defined range in the forged function is adaptive to the observed dataset.

## 4 EXPERIMENTS

### 4.1 SETUP

We evaluated the performance on CIFAR10, CIFAR100, and ImageNet datasets under the white-box scenario with an $L_\infty$ norm. To ensure comparability of results, we assessed robustness using AutoAttack (Croce & Hein, 2020), For CIFAR10 and CIFAR100 datasets, $\epsilon$ is set to $8/255$, while for the ImageNet dataset, $\epsilon$ is set to $4/255$. The model weights are publicly accessible from Robust-Bench. The ablation study involves exploring the selection of the optimal $c^r$, the combination of

---

**Algorithm 1** Forged Function

---

1: **require**: Input **x**, Mode $m$, Hyper-parameter $c^r$
2: **if** $m$ is tracking mode **then**
3:    $b = \max(b, |\mathbf{x}|)$
4: **else**
5:    $\mathcal{M} = \{x|\mathrm{abs}(x) \leq c^r b\}$
6:   **for all** $s \in \mathcal{M}$ **do**
7:     $s = 0$
8:   **end for**
9: **end if**
10: **return x**

---

Table 1: The results of top-3 competitors on Robustbench.

(a) CIFAR10 dataset

| # | Method | $\mathrm{acc_{nat}}$ | $\mathrm{acc_{AA}}$ |
|---|--------|---------|--------|
| * | Wang et al. (2023) + Ours | 93.20 | 71.70 |
| 1 | Peng et al. (2023) | 93.27 | 71.07 |
| 2 | Wang et al. (2023) | 93.25 | 70.69 |
| 3 | Bai et al. (2024) | 95.19 | 69.71 |

(b) CIFAR100 dataset

| # | Method | $\mathrm{acc_{nat}}$ | $\mathrm{acc_{AA}}$ |
|---|--------|---------|--------|
| * | Wang et al. (2023) + Ours | 74.97 | 44.00 |
| 1 | Wang et al. (2023) | 75.22 | 42.67 |
| 2 | Bai et al. (2024) | 83.08 | 41.80 |
| 3 | Cui et al. (2023) | 73.85 | 39.18 |

various models trained from different techniques, the verification of gradient masking, and assessments of certified adversarial robustness via randomized smoothing. Due to the page limit, the full experimental results of the ablation study are listed in Appendix A.

## 4.2 WHITE-BOX EVALUATION

### 4.2.1 PERFORMANCE ANALYSIS ON CIFAR10 AND CIFAR100 DATASETS

The model used in this study is based on WRN-70-16 architecture with SiLU function while generative data were involved during the training phase. The value of $c_i^{\mathrm{th}}$ is obtained by feeding all images from the training set without any augmentation, and $c^r$ was set to $2^{-8}$ for this experiment. Tables 1a and 1b summarize the top-3 competitors on Robustbench for CIFAR10 and CIFAR100 datasets, respectively, where # represent the rankings, our results are marked by the asterisk (*), $\mathrm{acc_{nat}}$ and $\mathrm{acc_{AA}}$ denote the accuracy against clean data and adversairal examples generated by AutoAttack, respectively.

As can be seen, our method combined with WRN-70-16 with SiLU function gains improvement in robustness by at least 0.9% and achieves the best results on Robustbench for both datasets. Nevertheless, standard accuracy ($\mathrm{acc_{nat}}$) is decreased. Many factors might affect the results. For example, the single additional hyper-parameter introduced in this study might not provide sufficient granularity to fit all layers in the target model.

### 4.2.2 PERFORMANCE ANALYSIS ON IMAGENET DATASET

In this experiment, we utilized the Swin (Liu et al., 2021) model architecture, a variant of transformers. However, scanning the approximately 1.2 million training images provided by the ImageNet dataset to determine the value of the hyper-parameter introduced in the forged function defined in (13) might take a long time. Alternatively, we randomly selected about 5,000 images as the observed images to determine the value of the hyper-parameter. Ideally, determining the optimal choice of $c^r$ requires conducting an ablation study to explore the relationship between the chosen $c^r$ and robust accuracy on a validation set. To accelerate this procedure, we first seek a value of $c^r$ with the highest standard accuracy. The candidate values are selected in a small range centered around this value.

Tables 2 lists the top-3 competitors on Robustbench for ImageNet dataset, including ranking, architecture, standard accuracy, and robust accuracy against AutoAttack. The experimental results demonstrate that the Swin-L model with GELU combined with our method can obtain improve-

Table 2: The results of top-3 competitors for ImageNet dataset on Robustbench.

| # | Method | Architecutre | $acc_{nat}$ | $acc_{AA}$ |
|---|---|---|---|---|
| * | Liu et al. (2023) + Ours | Swin-L | 78.88 | 60.04 |
| 1 | Liu et al. (2023) | Swin-L | 78.92 | 59.56 |
| 2 | Bai et al. (2024) | ConvNeXtV2-L + Swin-L | 81.48 | 58.50 |
| 3 | Liu et al. (2023) | ConvNeXt-L | 78.02 | 58.48 |

ments in robust accuracy and achieve the best result while standard accuracy has a tiny drop. This finding verifies that our method can be applied to both convolutional and fully connected layers.

### 4.2.3 COMBINATION WITH VARIOUS MODELS

The experiment aims to assess the generability of the proposed function on adversarial trained models with identical architecture but from various training strategies and to evaluate the potential cost reduction of adversarial training. We integrated the proposed approach with partial models selected from RobustBench, whose weights are obtained directly from the official without any modifications, and also included a model trained by TRADES (Zhang et al., 2019) as a baseline for CIFAR10 dataset. The selected models were trained using different techniques, such as adding perturbations in internal layers, retrieving information using knowledge distillation, reducing inefficient training data, or involving additional images from generated models or another dataset. Except for the model used in RST-WAP, which is WRN-28-10, the model architecture we utilized is WRN-34-10 with ReLU, as it is the most popular network on the RobustBench leaderboard (Croce et al., 2020).

For the white-box evaluation, the value of $c^r$ is set to $2^{-7}$. Table 3a and 3b present standard and robust accuracy of models integrated with our method for CIFAR10 and CIFAR100 dataset, respectively. In these tables, the column *Original* indicates the original results reported by RobustBench, and the column *Original+Ours* demonstrates the results of the proposed method. As indicated in these tables, for CIFAR10 dataset, the proposed method enhances robust accuracy by more than 2% for RST-AWP, DefEAT, and LTD models, while other models receive approximately 1 to 1.5% improvement in robustness. Similarly, for CIFAR100 dataset, these models meet at least a 1% increase in robustness. The empirical results prove that the resilience of existing models against adversarial attacks can be improved by Lemma 2. We believe that the proposed solution is general as it achieves great success in models incorporating different training techniques.

Another advantage of the proposed method that we would like to highlight is that the cost of our approach can almost be ignored compared to the cost of adversarial training as the cost involves only a single pass scan of a set of images to determine the hyper-parameter $c^{th}$. This implies that these models can enhance robustness for free. Specifically, LefEAT can achieve a robust accuracy of 57.30% by removing inefficient training data. By combining LefEAT model with our approach, a robust accuracy of 59.55% can be achieved, which is comparable to RST-AWP (60.04%). However, RST-AWP introduces more images from another dataset, resulting in a higher cost in each epoch. Similarly, for the CIFAR100 dataset, DefEAT with our proposed method achieves a robust accuracy of 32.11%, which is better than EffAug (31.85%), which involves more complex data augmentation during the training stage. This aligns with the suggestion by DefEAT that some data can be removed without hurting robustness. Holistically, we believe that our approach might provide a hint during the late phase of adversarial training to drop inefficient weights, resulting in further cost savings or enhanced resilience.

An interesting observation from these tables is that standard accuracy improves across all models for both CIFAR10 and CIFAR100 datasets. While this phenomenon is not directly explained by Lemma 2, we hypothesize that the output ranges of ReLU and our proposed functions are highly similar, allowing for the maintenance of accuracy on clean data.

### 4.2.4 GRADIENT MASKING VERIFICATION

Previous studies suggest that the resilience of models might be unintentionally overestimated (Athalye et al., 2018; Carlini et al., 2019). The proposed function in (9) suppresses values to zero if the condition is satisfied. One might argue that this property could unintentionally cause obfuscated

Table 3: Standard and robust accuracy of models integrated with our method.

(a) CIFAR10 dataset

| Method | Original | | Original+Ours | |
|---|---|---|---|---|
| | $acc_{nat}$ | $acc_{AA}$ | $acc_{nat}$ | $acc_{AA}$ |
| RST-AWP Wu et al. (2020) | 88.25 | 60.04 | 89.50 | 62.76 |
| DefEAT Chen & Lee (2024) | 86.54 | 57.30 | 87.40 | 59.55 |
| LTD Chen & Lee (2021) | 85.21 | 56.94 | 85.98 | 59.25 |
| AWP Wu et al. (2020) | 85.36 | 56.17 | 86.19 | 57.85 |
| TRADESZhang et al. (2019) | 85.34 | 52.86 | 85.78 | 53.80 |

(b) CIFAR100 dataset

| Method | Original | | Original+Ours | |
|---|---|---|---|---|
| | $acc_{nat}$ | $acc_{AA}$ | $acc_{nat}$ | $acc_{AA}$ |
| EffAug Addepalli et al. (2022) | 68.75 | 31.85 | 69.14 | 32.57 |
| DKLD Cui et al. (2023) | 64.08 | 31.65 | 64.26 | 32.58 |
| DefEAT Chen & Lee (2024) | 64.32 | 31.13 | 66.42 | 32.11 |
| LTD Chen & Lee (2021) | 64.07 | 30.59 | 64.29 | 31.95 |
| AWP Wu et al. (2020) | 60.38 | 28.86 | 60.63 | 29.72 |

gradients, resulting in gradient attacks being unable to efficiently produce adversarial examples. Therefore, to verify that the proposed method does not encounter the gradient masking issue, we should conduct more experiments from the following aspects:

1. White-box attacks should be better than black-box attacks.

2. Iterative attacks should have better performance than one-step attacks.

3. Robust accuracy should gradually decrease to zero when the radius of $\epsilon$-ball increase.

4. The modified model should defense against adversarial examples generated by the original models.

5. Certified robustness that conducted by random smoothing (Cohen et al., 2019).

AutoAttack has examined the first item, which involves three white-box attacks and one black-box attack. By comparing robust accuracy shown in Table 1a, 1b and 2, the models combined with the proposed method perform better robust accuracy than the original models. It indicates black-box attacks cannot produce more adversarial examples.

The full experimental results for the rest of the experiments can be found in Appendix B. The results demonstrate that the proposed algorithm does not violate any of the above rules and the certified robustness improves by our method across most settings. From the evidence, we believe that the proposed method does not encounter the gradient masking problem among different hyper-parameters and various models on CIFAR10 and CIFAR100 datasets.

## 4.3 COST ANALYSIS

For the CIFAR-10 and CIFAR-100 datasets, the total training time for AWP on a V100 GPU is approximately 20 hours. In contrast, other methods use larger models and incorporate additional data (Wang et al., 2023; Peng et al., 2023; Bai et al., 2024), leading to a total training time of over 200 hours. In comparison, scanning the entire CIFAR-10 or CIFAR-100 dataset, or partial training set on ImageNet dataset, takes less than 5 minutes on a V100 GPU.

Regarding the costs of hyperparameter search, as discussed in Section A.1, there are only three candidate parameters to evaluate performance. We can efficiently assess white-box performance using partial data from the training set, which significantly reduces the computational time. In practice, the total time for hyperparameter search is about 1-3 hours, depending on the model size

and the dataset used. This demonstrates that our approach incurs significantly lower computational overhead.

### 4.4 DISCUSSION

A common pitfall is the misconception that minimizing the magnitude of the Lipschitz constant necessarily leads to improved robustness. However, in an extreme scenario, if the weights of the linear layers are replaced with an identity matrix, the Lipschitz constant would equal 1. This change would completely alter the output distribution, resulting in zero natural accuracy and, therefore, no meaningful robustness to assess. This work demonstrates that by manipulating the input domain of a linear system, it is possible to obtain an equivalent matrix that produces the same output while having a Lipschitz constant that is equal to or less than the original.

While it is true that the Lipschitz constant can be reduced through certified training (Mao et al., 2024) or Lipschitz-constrained methods (Zühlke & Kudenko, 2024), these approaches often introduce additional regularization terms or more complex objectives, complicating the training of robust models. In contrast, this work is specifically designed for model optimization during inference time, providing a cost-effective means to enhance robustness. Furthermore, this research offers valuable insights into identifying which weights can be eliminated with minimal impact on performance. This property could be integrated into existing training frameworks to further enhance robustness.

The overall procedure of the proposed algorithm shares many similarities with pruning techniques (He & Xiao, 2023). However, we would like to emphasize that the proposals are distinctly different. Pruning primarily aims to create a highly sparse model that accelerates inference times or reduces the model size for deployment on edge devices, often without considering robustness. In particular, when the pruned model exhibits extremely high sparsity without applying re-training or fine-tuning, natural accuracy drops significantly, implicitly indicating that condition (11) does not hold and that Lemma 2 is not applicable in this case. On the other hand, when the pruned model has low sparsity, only a small proportion of weights with values close to zero are eliminated, resulting in insignificant adjustments to the center and radius of the disk. The proposed algorithm, on the other hand, can identify crucial elements with minimal cost. We believe that investigating the impact of various pruning techniques, such as iterative or post-training methods, on robustness, or combining these techniques with our proposed approach, represents a valuable direction for future research.

## 5 CONCLUSION

In this paper, we recap how robustness is certified by the theorem of Lipschitz continuity. We introduce the concept of the empirical Lipschitz constant, which minimizes the influence of the space not drawn from real data, resulting in a precise estimation of the robustness of the corresponding observed data. We prove that by remapping the input domain of a specific layer to a constrained range, the Lipschitz constant can be shrunk, leading to better robustness. The proposed function introduces only one parameter, the value of which can be determined by scanning the training data once, without re-training or fine-tuning. Compared with adversarial training, the proposed method is almost cost-free. The experimental results suggest that our method can be combined with various existing methods and achieve robustness improvements, and no gradient masking occurs in our algorithm. Furthermore, our method can achieve the best robust accuracy for CIFAR10, CIFAR100, and ImageNet datasets on the RobustBench leaderboard.

Numerous future directions merit exploration. Firstly, due to the property of maximization, the proposed function might easily be influenced by outliers. Designing a better function is an interesting research topic. Secondly, exploring the combination with various activation functions, different model architectures or large-scale datasets would be beneficial. Lastly, it is worth investigating to understand the theoretical reasons why our proposed function improves standard accuracy.

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

## A  ABLATION STUDY

### A.1  HYPER-PARAMETER SELECTION

This experiment investigates how the choice of hyper-parameter $c^r$ influences standard accuracy and robust accuracy. Since most models are represented in 16 bit format, and the widths of fraction bit for FP16 format defined by IEEE-754 standard and BFloat are 10 and 7 bits, respectively, truncated errors might easily occur when performing addition on two numbers with a magnitude difference of $2^8$ or higher. On the other hand, when $c^r$ is set to $2^{-5}$, all models experience a significant drop in standard accuracy, and there is meaningless in evaluating robustness at this configuration. We suggest that the candidates of $c^r$ are $2^{-8}$, $2^{-7}$ and $2^{-6}$.

The results on CIFAR10 and CIFAR100 are presented in Table 4 and Table 5, respectively. More-over, the results of accuracy against CW attack on $L_\infty$ norm for CIFAR10 and CIFAR100 datasets are presented in Tables 6a and 6b, respectively. As can be seen, when $c^r$ is set to $2^{-8}$, all models achieve better standard accuracy and robust accuracy. Additionally, the results for all models with $c^r = 2^{-7}$ are surpassed by those when $c^r$ is set to $2^{-8}$. Robust accuracy can be further enhanced by setting $2^{-6}$, while standard accuracy might drop compared to the original. The results suggest that $c^r = 2^{-7}$ is a solution that balances standard accuracy and robustness. Nevertheless, when robustness is a major concern, $c^r = 2^{-6}$ is a better choice.

Intuitively, we expect that standard accuracy gradually decreases when the value of $c^r$ increases. The phenomenon can be observed when $c^r$ is $2^{-6}$ or higher but two counterexamples are reported in the ablation study when setting $c^r$ to $2^{-7}$ and $2^{-8}$. A possible explanation is that the optimizer becomes stuck in a saddle area, as ReLU is non-differentiable at the zero point. This might cause the gradient direction to become stuck in an oscillation when values are close to zero. By shifting those values to zero, antagonistic effects among different feature maps, filters, or channels are accidentally mitigated. However, further investigation and evidence are needed to support this conjecture.

We argue that any function that satisfies the conditions defined in (11) can shrink the largest eigen-value. There might be another function that can perform better than the proposed one. Besides, the hyper-parameter is determined by choosing the maximum value appearing in the dataset.

## B  FULL EXPERIMENTAL RESULTS OF GRADIENT MASKING VERIFICATION

Table 7a and 7b present the robust accuracy against adversarial examples generated by the origi-nal models on CIFAR10 and CIFAR100 datasets, respectively. As observed, none of the models showed lower robust accuracy than the original model. It indicates that adversarial examples can be efficiently crafted by utilizing the gradients from the victim models.

Table 8 and 9 presents the robust accuracy against FGSM and PGD attacks among different radii of the $\epsilon$-ball on the CIFAR10 and CIFAR100 datasets, respectively. As observed, the robust accuracy

Table 4: Ablation study of selecting optimal $c^r$ for CIFAR10 dataset.

| Method | RobustBench | | $c^r = 2^{-8}$ | | $c^r = 2^{-7}$ | | $c^r = 2^{-6}$ | |
|---|---|---|---|---|---|---|---|---|
| | $\text{acc}_{\text{nat}}$ | $\text{acc}_{\text{AA}}$ | $\text{acc}_{\text{nat}}$ | $\text{acc}_{\text{AA}}$ | $\text{acc}_{\text{nat}}$ | $\text{acc}_{\text{AA}}$ | $\text{acc}_{\text{nat}}$ | $\text{acc}_{\text{AA}}$ |
| RST-AWP | 88.25 | 60.04 | 88.82 | 60.96 | 89.50 | 62.76 | 87.88 | 61.96 |
| DefEAT | 86.54 | 57.30 | 86.88 | 57.81 | 87.40 | 59.55 | 84.59 | 61.08 |
| LTD | 85.21 | 56.94 | 85.28 | 57.28 | 85.98 | 59.25 | 85.59 | 60.63 |
| AWP | 85.36 | 56.17 | 85.80 | 56.53 | 86.19 | 57.85 | 84.55 | 59.21 |
| TRADES | 85.34 | 52.86 | 85.57 | 52.97 | 85.78 | 53.80 | 85.49 | 55.37 |

Table 5: Ablation study of selecting optimal $c^r$ for CIFAR100 dataset.

| Method | RobustBench | | $c^r = 2^{-8}$ | | $c^r = 2^{-7}$ | | $c^r = 2^{-6}$ | |
|---|---|---|---|---|---|---|---|---|
| | $\text{acc}_{\text{nat}}$ | $\text{acc}_{\text{AA}}$ | $\text{acc}_{\text{nat}}$ | $\text{acc}_{\text{AA}}$ | $\text{acc}_{\text{nat}}$ | $\text{acc}_{\text{AA}}$ | $\text{acc}_{\text{nat}}$ | $\text{acc}_{\text{AA}}$ |
| EffAug | 68.75 | 31.85 | 68.81 | 32.00 | 69.14 | 32.57 | 68.44 | 33.64 |
| DKLD | 64.08 | 31.65 | 64.10 | 31.77 | 64.26 | 32.58 | 63.50 | 33.87 |
| DefEAT | 65.89 | 30.57 | 66.12 | 31.11 | 66.42 | 32.46 | 65.06 | 34.07 |
| LTD | 64.07 | 30.59 | 64.29 | 31.13 | 64.29 | 31.95 | 64.18 | 34.04 |
| AWP | 60.38 | 28.86 | 60.18 | 29.10 | 60.63 | 29.72 | 60.71 | 30.82 |

Table 6: The robust accuracy against CW attack on $L_\infty$ norm.

(a) CIFAR10 dataset

| Method | Origin | $c^r$ | | |
|---|---|---|---|---|
| | | $2^{-8}$ | $2^{-7}$ | $2^{-6}$ |
| RST-AWP | 58.98 | 61.84 | 68.24 | 80.92 |
| DefEAT | 56.92 | 58.02 | 61.06 | 65.56 |
| LTD | 58.12 | 58.56 | 60.50 | 64.86 |
| AWP | 56.84 | 57.34 | 60.58 | 66.50 |
| TRADES | 56.10 | 56.52 | 58.18 | 63.62 |

(b) CIFAR100 dataset

| Method | Origin | $c^r$ | | |
|---|---|---|---|---|
| | | $2^{-8}$ | $2^{-7}$ | $2^{-6}$ |
| EffAug | 37.40 | 37.70 | 38.70 | 43.00 |
| DKLD | 37.50 | 38.06 | 39.38 | 44.20 |
| DefEAT | 36.90 | 37.56 | 39.82 | 44.30 |
| LTD | 36.66 | 37.32 | 38.86 | 43.44 |
| AWP | 34.56 | 35.20 | 35.94 | 40.40 |

against FGSM, a one-step attack, is always higher than the robust accuracy against PGD, an iterative attack. This implies that the gradient is reliable, allowing the PGD attack to adjust the gradient direction multiple times to find adversarial examples. Additionally, we observe that the robust accuracy against PGD attacks for all models gradually decreases to zero as the radius of the $\epsilon$-ball increases. This indicates that the quality of gradients is preserved, enabling PGD attacks to move the gradient toward examples not in the observed distribution.

Figure 3 illustrates the certified robustness achieved by random smoothing for various models on the CIFAR10 dataset, where *Original* refers to the certified robustness of the original model, while *Ours* denotes the robustness of the model combined with the proposed method. As can be seen, our method brings slight improvements in robustness, except for the AWP model. These results demonstrate that our algorithm does not suffer from the gradient masking issue. However, the empirical Lipschitz constant is derived from the observed data. As the input distribution drawn from random smoothing and the observed data might have discrepancies, this could result in fluctuations in robustness.

## C  QUANTITATIVE ANALYSIS OF EMPIRICAL LIPSCHITZ CONSTANT

As mentioned in Definition 1, the Lipschitz constant in this paper is estimated based on the observed data. The sparsity of the forged vectors is a crucial factor influencing the magnitude of the Lipschitz constant for the corresponding layers, although it is not the only factor. Figure 4 illustrates the average proportion of pruned activations, which varies depending on the location of each linear layer. FC1 and FC2 refer to the first and second fully connected layers in the MLP blocks, respectively.

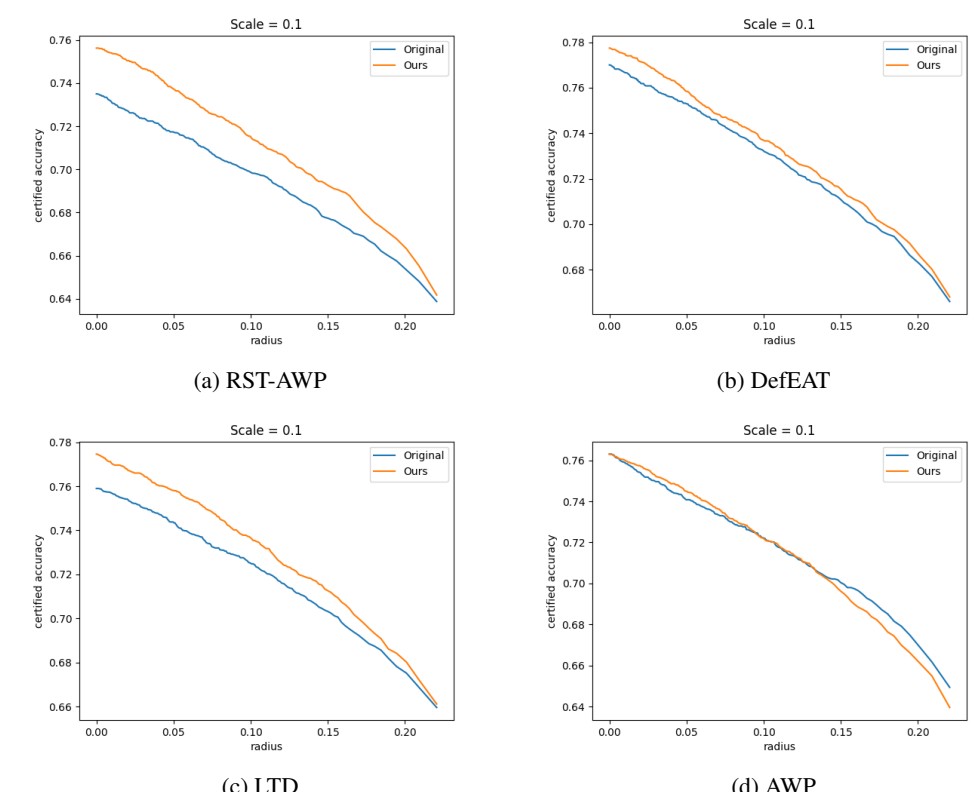

Figure 3: Certified robustness that conducted by random smoothing.

Table 7: The robust accuracy against adversarial examples generated by the original models.

(a) CIFAR10 dataset

| Method | Origin | $c^r$ | | |
| --- | --- | --- | --- | --- |
| | | $2^{-8}$ | $2^{-7}$ | $2^{-6}$ |
| RST-AWP | 60.04 | 62.10 | 65.10 | 70.53 |
| DefEAT | 57.30 | 58.37 | 60.39 | 66.10 |
| LTD | 56.94 | 58.71 | 61.63 | 66.47 |
| AWP | 56.17 | 57.49 | 59.74 | 65.58 |
| TRADES | 52.86 | 55.55 | 55.09 | 58.68 |

(b) CIFAR100 dataset

| Method | Origin | $c^r$ | | |
| --- | --- | --- | --- | --- |
| | | $2^{-8}$ | $2^{-7}$ | $2^{-6}$ |
| EffAug | 31.85 | 32.87 | 35.08 | 40.04 |
| DKLD | 31.65 | 32.91 | 35.04 | 40.58 |
| DefEAT | 30.57 | 31.82 | 33.94 | 40.67 |
| LTD | 30.59 | 32.05 | 34.07 | 39.11 |
| AWP | 28.86 | 29.88 | 32.18 | 36.67 |

As shown, the proportion of pruned activations is approximately 20% for the FC1 layers, except for the last layer. This suggests that the forged function alone cannot significantly reduce the magnitude of the Lipschitz constant.

On the other hand, the pruned rates for the FC2 layer range from 30% to 95%, depending on the layer's location. However, we emphasize that a higher pruning rate does not necessarily lead to a substantial reduction in the empirical Lipschitz constant. This is because, although the Lipschitz constant estimated from most of the data may decline, the empirical Lipschitz constant is based on the worst-case scenario across the entire observed dataset. Experimental results show that, after applying our method, the eigenvalue of the worst-case scenario is approximately 95% of the original eigenvalue.

Table 8: The robust accuracy against FGSM and PGD attacks among different radii of $\epsilon$-ball on CIFAR10 dataset.

| Method | $c^r$ | Attack | $\epsilon$ | | | | | | | |
| --- | --- | --- | --- | --- | --- | --- | --- | --- | --- | --- |
| | | | $\frac{1}{255}$ | $\frac{2}{255}$ | $\frac{4}{255}$ | $\frac{8}{255}$ | $\frac{16}{255}$ | $\frac{32}{255}$ | $\frac{64}{255}$ | $\frac{96}{255}$ |
| RST-AWP | $2^{-8}$ | FGSM | 88.28 | 86.94 | 83.80 | 75.12 | 57.23 | 34.04 | 18.80 | 19.03 |
| | | PGD | 86.78 | 84.47 | 79.03 | 66.03 | 34.02 | 2.01 | 0.01 | 0.0 |
| | $2^{-7}$ | FGSM | 89.46 | 88.28 | 85.64 | 77.62 | 60.60 | 35.91 | 19.39 | 20.68 |
| | | PGD | 88.03 | 85.88 | 80.89 | 69.24 | 38.27 | 3.08 | 0.1 | 0.0 |
| | $2^{-6}$ | FGSM | 87.70 | 86.63 | 84.38 | 77.91 | 60.93 | 33.48 | 16.12 | 18.81 |
| | | PGD | 86.38 | 84.69 | 81.19 | 73.72 | 52.24 | 11.89 | 0.19 | 0.0 |
| DefEAT | $2^{-8}$ | FGSM | 86.38 | 85.40 | 81.98 | 72.73 | 53.28 | 30.34 | 18.13 | 19.65 |
| | | PGD | 84.52 | 82.07 | 76.51 | 63.71 | 33.87 | 1.76 | 0.0 | 0.0 |
| | $2^{-7}$ | FGSM | 86.69 | 85.70 | 83.05 | 74.35 | 56.36 | 32.04 | 18.88 | 21.40 |
| | | PGD | 85.11 | 82.87 | 78.00 | 66.54 | 38.70 | 3.12 | 0.0 | 0.0 |
| | $2^{-6}$ | FGSM | 84.14 | 83.57 | 81.03 | 74.63 | 57.67 | 28.11 | 13.17 | 19.38 |
| | | PGD | 82.96 | 81.35 | 77.37 | 69.52 | 50.70 | 10.05 | 0.2 | 0.0 |
| LTD | $2^{-8}$ | FGSM | 84.94 | 83.87 | 81.15 | 72.80 | 55.45 | 33.06 | 18.44 | 17.48 |
| | | PGD | 83.13 | 80.68 | 75.53 | 63.52 | 34.81 | 2.64 | 0.0 | 0.0 |
| | $2^{-7}$ | FGSM | 85.48 | 84.67 | 82.24 | 74.10 | 57.41 | 35.53 | 17.57 | 17.50 |
| | | PGD | 83.88 | 81.88 | 77.00 | 65.38 | 28.57 | 3.95 | 0.0 | 0.0 |
| | $2^{-6}$ | FGSM | 85.06 | 84.28 | 82.21 | 75.77 | 60.79 | 34.39 | 14.34 | 15.42 |
| | | PGD | 83.91 | 82.00 | 78.33 | 69.82 | 49.95 | 11.63 | 0.21 | 0.0 |
| AWP | $2^{-8}$ | FGSM | 85.11 | 83.90 | 80.68 | 71.28 | 53.78 | 33.21 | 20.86 | 19.61 |
| | | PGD | 83.34 | 80.34 | 75.08 | 61.53 | 30.50 | 1.89 | 0.03 | 0.0 |
| | $2^{-7}$ | FGSM | 85.50 | 84.68 | 81.75 | 73.56 | 57.19 | 35.50 | 20.63 | 20.16 |
| | | PGD | 83.94 | 81.62 | 76.34 | 65.57 | 34.16 | 2.89 | 0.02 | 0.0 |
| | $2^{-6}$ | FGSM | 83.87 | 83.19 | 81.08 | 74.97 | 61.49 | 35.13 | 16.12 | 18.69 |
| | | PGD | 83.00 | 81.42 | 78.13 | 71.50 | 54.95 | 14.78 | 0.41 | 0.0 |
| TRADES | $2^{-8}$ | FGSM | 84.74 | 83.51 | 70.58 | 70.50 | 54.23 | 36.53 | 23.97 | 23.46 |
| | | PGD | 82.62 | 79.72 | 72.81 | 57.30 | 24.21 | 1.21 | 0.01 | 0.0 |
| | $2^{-7}$ | FGSM | 85.00 | 84.02 | 80.57 | 71.61 | 55.94 | 25.67 | 22.31 | 22.51 |
| | | PGD | 82.96 | 80.31 | 73.75 | 58.91 | 26.22 | 1.31 | 0.02 | 0.0 |
| | $2^{-6}$ | FGSM | 85.05 | 84.19 | 81.40 | 75.07 | 60.24 | 36.99 | 21.81 | 21.21 |
| | | PGD | 83.23 | 81.33 | 75.81 | 64.56 | 36.76 | 3.37 | 0.02 | 0.0 |

Table 9: The robust accuracy against FGSM and PGD attacks among different radii of $\epsilon$-ball on CIFAR100 dataset.

| Method | $c^r$ | Attack | $\frac{1}{255}$ | $\frac{2}{255}$ | $\frac{4}{255}$ | $\frac{8}{255}$ | $\frac{16}{255}$ | $\frac{32}{255}$ | $\frac{64}{255}$ | $\frac{96}{255}$ |
|---|---|---|---|---|---|---|---|---|---|---|
| | | | | | | $\epsilon$ | | | | |
| EffAug | $2^{-8}$ | FGSM | 68.02 | 65.96 | 60.36 | 49.65 | 33.90 | 17.39 | 7.11 | 5.77 |
| | | PGD | 64.92 | 61.04 | 52.82 | 39.37 | 17.53 | 1.81 | 0.0 | 0.0 |
| | $2^{-7}$ | FGSM | 68.41 | 66.56 | 61.84 | 51.83 | 36.59 | 18.34 | 7.02 | 6.70 |
| | | PGD | 65.66 | 62.02 | 54.58 | 41.55 | 19.70 | 2.26 | 0.0 | 0.0 |
| | $2^{-6}$ | FGSM | 67.84 | 67.25 | 64.65 | 57.97 | 44.23 | 22.20 | 8.26 | 8.85 |
| | | PGD | 66.38 | 64.20 | 59.63 | 50.89 | 33.98 | 7.27 | 0.17 | 0.0 |
| DKLD | $2^{-8}$ | FGSM | 63.47 | 61.94 | 58.15 | 48.86 | 34.39 | 17.73 | 6.26 | 3.66 |
| | | PGD | 60.71 | 57.14 | 50.44 | 38.14 | 17.38 | 1.97 | 0.0 | 0.0 |
| | $2^{-7}$ | FGSM | 63.55 | 62.31 | 58.65 | 50.37 | 36.49 | 18.69 | 6.36 | 4.29 |
| | | PGD | 61.06 | 57.84 | 51.51 | 39.99 | 19.71 | 2.37 | 0.0 | 0.0 |
| | $2^{-6}$ | FGSM | 63.26 | 62.77 | 60.41 | 55.18 | 43.86 | 21.17 | 5.50 | 4.97 |
| | | PGD | 61.67 | 59.62 | 55.83 | 48.06 | 33.51 | 7.55 | 0.17 | 0.0 |
| DefEAT | $2^{-8}$ | FGSM | 65.57 | 64.36 | 59.88 | 49.38 | 32.48 | 15.38 | 5.50 | 3.30 |
| | | PGD | 62.39 | 58.96 | 51.85 | 38.59 | 17.18 | 1.45 | 0.0 | 0.0 |
| | $2^{-7}$ | FGSM | 65.97 | 64.96 | 60.67 | 51.44 | 34.84 | 16.36 | 5.33 | 3.96 |
| | | PGD | 62.94 | 59.83 | 52.98 | 41.05 | 20.07 | 2.02 | 0.0 | 0.0 |
| | $2^{-6}$ | FGSM | 64.69 | 63.58 | 61.31 | 54.47 | 40.06 | 16.91 | 4.64 | 4.46 |
| | | PGD | 62.85 | 60.50 | 56.29 | 47.54 | 30.76 | 5.84 | 0.09 | 0.0 |
| LTD | $2^{-8}$ | FGSM | 63.59 | 62.35 | 58.10 | 48.86 | 33.11 | 16.78 | 5.92 | 3.20 |
| | | PGD | 61.06 | 57.65 | 50.70 | 38.21 | 18.21 | 1.98 | 0.0 | 0.0 |
| | $2^{-7}$ | FGSM | 64.05 | 62.87 | 59.02 | 50.16 | 34.90 | 17.27 | 5.54 | 3.11 |
| | | PGD | 61.51 | 58.32 | 51.84 | 39.89 | 20.32 | 2.26 | 0.0 | 0.0 |
| | $2^{-6}$ | FGSM | 63.62 | 62.98 | 60.96 | 54.96 | 41.51 | 19.78 | 4.69 | 3.12 |
| | | PGD | 61.90 | 59.68 | 55.23 | 46.49 | 29.13 | 5.74 | 0.05 | 0.0 |
| AWP | $2^{-8}$ | FGSM | 59.77 | 58.06 | 54.21 | 45.54 | 30.98 | 16.69 | 6.49 | 3.97 |
| | | PGD | 56.72 | 52.92 | 46.53 | 34.90 | 16.03 | 2.11 | 0.0 | 0.0 |
| | $2^{-7}$ | FGSM | 60.00 | 58.52 | 54.93 | 46.65 | 32.66 | 17.42 | 6.04 | 3.60 |
| | | PGD | 57.19 | 53.75 | 47.46 | 36.22 | 17.48 | 2.58 | 0.0 | 0.0 |
| | $2^{-6}$ | FGSM | 60.20 | 59.71 | 57.47 | 52.05 | 39.78 | 21.10 | 5.70 | 3.90 |
| | | PGD | 58.19 | 55.66 | 50.91 | 42.37 | 25.53 | 5.68 | 0.09 | 0.0 |

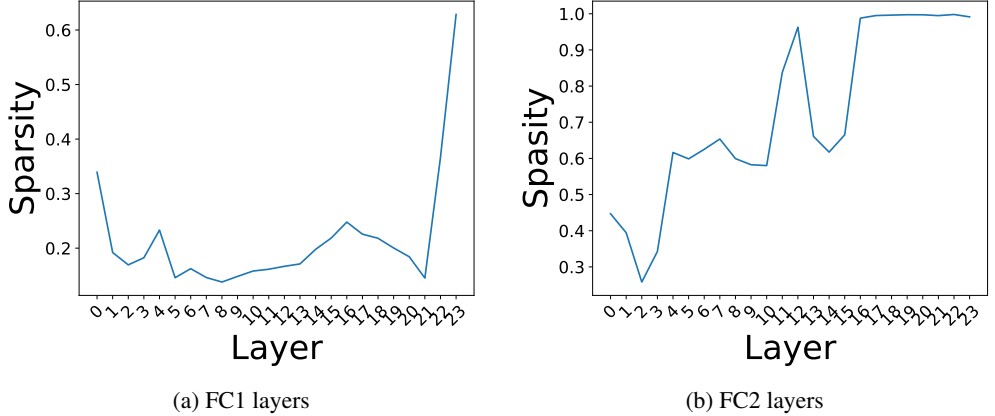

(a) FC1 layers          (b) FC2 layers

Figure 4: The average proportion of pruned activations depends on the location of each linear layer. FC1 and FC2 refer to the first and second fully connected layers in MLP blocks, respectively.