# OpenReview forum: "Data-Driven Lipschitz Continuity: A Cost-Effective Approach to Improve Adversarial Robustness"
_ICLR.cc/2025/Conference — Submitted to ICLR 2025_

### Official Review · Reviewer_Erqp · 2024-10-31

**Soundness:** 2
**Presentation:** 2
**Contribution:** 2
**Rating:** 3
**Confidence:** 3

**Summary:**

This work aims to enhance the adversarial robustness of DNNs by reducing the Lipschitz constant. Unlike existing adversarially trained models, this method is a cost-free method to enhance adversarial robustness by introducing a forged function in the DNNs. Experimental results on CIFAR10, CIFAR100, and ImageNet datasets have demonstrated the proposed method outperforms the adversarial training method.

**Strengths:**

1. The idea of enhancing the adversarial robustness by reducing the Lipschitz constant is reasonable and straightforward.

2. This paper is well structured.

**Weaknesses:**

1. The concept of utilizing the Lipschitz constant to certify the adversarial robustness of DNNs has been explored in many previous works [1-3]. This Lipschitz constant is utilized to provide a provable adversarial robustness. However, this paper does not discuss the main differences and improvements from those provable adversarial defense methods.

2. The proposed forged function seems to be a variation of ReLU, just with a different threshold. Additionally, as illustrated in Figure 2, the Forged function is concatenated with ReLU. How does this differ from simply applying a shifted ReLU?

3. Given Equation (3), if the model becomes larger and more complex, would this increase the Lipschitz constant, potentially worsening the model's robustness?


[1] Certified adversarial robustness via randomized smoothing, In ICML2019
[2] Provably robust deep learning via adversarially trained smoothed classifiers, In Neurips 2019
[3] Scaling provable adversarial defenses, In Neurips 2018.

**Questions:**

See weakness.

---

> ### Author Response · Authors · 2024-11-15
> **Rebuttal by Authors**
>
> **[Q5-1]** The concept of utilizing the Lipschitz constant to certify the adversarial robustness of DNNs has been explored in many previous works [1-3]. This Lipschitz constant is utilized to provide a provable adversarial robustness. However, this paper does not discuss the main differences and improvements from those provable adversarial defense methods.
>
> **[A5-1]** We would like to clarify that the proposed algorithm significantly differs from existing Lipschitz constrained methods. The existing Lipschitz constrained methods primarily focus on designing an objective to minimize the magnitude of the Lipschitz constant of the learned weights during the training phase. In contrast, our method seeks a constrained set and remaps the internal features that fed into a linear system into this set. As indicated by Lemma 2, the Lipschitz constant of the refined model is non-increasing.
>
> It is important to highlight that our method targets different aspects of the learning process. Moreover, our approach offers the advantage of identifying the constrained set by scanning the entire or partial training set without the need for retraining the models, which is a significant benefit over existing methods. This distinct strategy and its practical benefits contribute to advancing the literature in this field.
>
> **[Q5-2]** The proposed forged function seems to be a variation of ReLU, just with a different threshold. Additionally, as illustrated in Figure 2, the Forged function is concatenated with ReLU. How does this differ from simply applying a shifted ReLU?
>
> **[A5-2]** While the forged function shares some similarities with ReLU, there are key differences. First, our forged function introduces an additional parameter, $c^r$, which can be adjusted for each layer. This allows more fine-grained control over the behavior of the function at different stages of the network. Second, the forged function is not limited to just being a shifted ReLU; it can also be integrated with other activation functions, such as SiLU and GELU, offering greater flexibility and expressiveness. These differences allow the forged function to better adapt to the specific needs of the model, contributing to improved robustness.
>
> **[Q5-3]** Given Equation (3), if the model becomes larger and more complex, would this increase the Lipschitz constant, potentially worsening the model's robustness?
>
> **[A5-3]** As mentioned in Section 4.3, a common misconception is that minimizing the magnitude of the Lipschitz constant necessarily leads to improved robustness. However, in an extreme case, if the weights of the linear layers are replaced with an identity matrix, the Lipschitz constant would be equal to 1. This change would drastically alter the output distribution, resulting in zero natural accuracy and, therefore, no meaningful robustness to assess.
>
> It is important to note that the Lipschitz constant alone is not a sufficient measure for comparing models. Our work demonstrates that by manipulating the input domain of a linear system, it is possible to obtain an equivalent matrix that produces the same output while having a Lipschitz constant that is less than or equal to the original. This manipulation enables improved robustness without compromising the model’s performance.

---

### Official Review · Reviewer_q4ED · 2024-11-01

**Soundness:** 2
**Presentation:** 3
**Contribution:** 2
**Rating:** 5
**Confidence:** 4

**Summary:**

The paper proposes a model agnostic method for improving the robustness of the network. Essentially, the method prune the activation of small values of the pretrained models. The pruning threshold is determined by the data and a hyperparameter. The paper performs a series experiment on different kinds of adversarially-trained models and shows that the proposed method can improve the robust accuracy by about 0.5%-2% while keeping the accuracy on the natural images. The paper discuss the possibility of gradient masking. However, the experiments cannot fully eliminate such possibility.

**Strengths:**

1. The paper proposes a model agnostic method that can be easily applied to various robust models and improve the robustness of these method. Such methods have good potential for wide applications.

2. While the proposed method is simple, the pruning technique shows to improve the robust accuracy of various models. Considering the newly introduced hyperparameters are limited, the improvement is far above the margin of errors.

**Weaknesses:**

1. While the paper discuss the possibility of gradient masking, the experiment is not enough to completely rule out such possibilities. Specially, the paper uses forging function to cut specific activation. Therefore, attacking methods like BPDA should be combined into AutoAttack and the robust accuracy under such attacks should be reported.

2. The motivation of the paper is not fully elaborated in the experiment. For example, whether the empirical Lipschitz has truly been lowered by such method should be reported.

3. Despite the simplicity the method, the novelty of  the proposed method is limited.

**Questions:**

1. As $c^r$ is very small, I wonder the proportion of activation that be pruned in the experiment.

2. As the proposed method change the activation even in the natural images, why the natural accuracy is improved in some cases?

---

> ### Author Response · Authors · 2024-11-15
> **Rebuttal by Authors**
>
> **[Q4-1]** While the paper discuss the possibility of gradient masking, the experiment is not enough to completely rule out such possibilities. Specially, the paper uses forging function to cut specific activation. Therefore, attacking methods like BPDA should be combined into AutoAttack and the robust accuracy under such attacks should be reported.
>
> **[A4-1]** BPDA is one approach that can be used to assess potential gradient obfuscation. The core idea behind BPDA is to approximate the gradient using a differentiable approximation [r4-1]. In the case of the forged function, when elements close to zero are replaced with zero, this can lead to a zero gradient. BPDA compensates for this by replacing the zero gradient with the small values. However, when $c^r$ is larger, the range of values being masked becomes broader, and BPDA becomes less practical. The results of transfer attacks, as shown in Table 7, align with this observation, indicating that the method remains robust under these attack scenarios.
>
> While we acknowledge that both attack and defense methods will continue to evolve, we believe our approach is highly effective against existing attack methods. Currently, there is no evidence suggesting that our method introduces gradient obfuscation that could make it vulnerable to untested attacks. Therefore, we are confident in the robustness of our method against current attack techniques.
>
> **[Q4-2]** The motivation of the paper is not fully elaborated in the experiment. For example, whether the empirical Lipschitz has truly been lowered by such method should be reported.
>
> **[A4-2]** We appreciate the reviewer’s suggestion. We conducted an experiment to examine the behavior of the eigenvalues after applying the proposed method. The experimental results indicate that the average eigenvalue after applying the method is approximately 95% of the original eigenvalue. We will include this experiment in the appendix to provide further empirical support for our approach.
>
> **[Q4-3]** Despite the simplicity the method, the novelty of the proposed method is limited.
>
> **[A4-3]** We would like to emphasize that the primary motivation behind our work is to address the high cost of adversarial training, which is often unaffordable for many users. Our method offers a cost-effective solution for inference-time model optimization, providing a way to enhance robustness without requiring extensive computational resources. We believe this makes our method an important contribution, particularly for users who cannot afford adversarial training on large-scale datasets.
>
> **[Q4-4]** As $c^r$ is very small, I wonder the proportion of activation that be pruned in the experiment.
>
> **[A4-4]** The proportion of pruned activations depends on the location of each linear layer. Generally, for the first fully connected layers in MLP blocks (fc1), around 10-20% of the activations can be pruned. In contrast, the second fully connected layers in MLP blocks (fc2) can achieve a sparsity of around 50-60%. This varies depending on the architecture, but these are typical ranges observed in our experiments. I will consider including the discussion in the revised version.
>
> **[Q4-5]** As the proposed method change the activation even in the natural images, why the natural accuracy is improved in some cases?
>
> **[A4-5]** We acknowledge that this is an interesting and somewhat unexpected outcome. Currently, we do not have a definitive explanation for why natural accuracy improves in some cases. We have observed this phenomenon and have noted it as a topic for future investigation in the discussion section. Further research could help uncover the underlying mechanisms contributing to this behavior.
>
> References:
>
> [r4-1] Athalye, A., Carlini, N., & Wagner, D. (2018, July). Obfuscated gradients give a false sense of security: Circumventing defenses to adversarial examples. In International conference on machine learning (pp. 274-283). PMLR.

---

> ### Comment · Reviewer_q4ED · 2024-12-02
> **Response to authors**
>
> Thanks for the response of the authors. I am not persuaded by the authors as [A4-1] as Eqn (9) is clearly a operation that stops gradient. The gradient of identity transform is 1 while the gradient of the degraded value is 0. This actually corresponds to the gradient masking behavior that BPDA wants to resolve.
>
> I will maintain my score unchanged.

---

### Official Review · Reviewer_exHp · 2024-11-03

**Soundness:** 3
**Presentation:** 3
**Contribution:** 3
**Rating:** 6
**Confidence:** 4

**Summary:**

The author proposes a plug-play method to enhance the adversarial robustness of deep neural networks by focusing on Lipschitz continuity. This work reduces the Lipschitz constant by remapping the input domain to a constrained range, thus improving robustness without requiring retraining. This approach is almost cost-free, and experimental results combined with the previous work show the effectiveness of the proposed method.

**Strengths:**

1.	The method is highly cost-effective as it does not require model retraining or additional data. Moreover, due to its plug-and-play nature, the proposed method can be easily integrated with existing algorithms.
2.	The authors provide detailed theoretical insights into the relationship between the Lipschitz constant and adversarial robustness.
3.	The paper is well-organized and clearly presented, with extensive experiments across diverse datasets that validate the robustness of the proposed method.

**Weaknesses:**

1.	The experimental results, along with the authors' own analysis, indicate that the proposed method may reduce the classification accuracy on clean samples in some cases. While the authors discuss possible reasons, the paper could further analyze this trade-off and explore strategies to mitigate accuracy loss.
2.	The choice of parameter $c^r$ is crucial to performance, but finding a single $c^r$ that performs well across all tasks is challenging based on the results presented. For example, $c^r=2^{-6}$ performs well on most tasks, while $c^r=2^{-7}$ works better for RST-AWP in Table 4. Adjusting this parameter for new tasks could be time-consuming.

**Questions:**

1.	In the evaluation using $acc_{AA}$, are the adversarial examples generated using the original neural network architecture, or using the architecture with the proposed forged function?
2.	Can this method be extended to other tasks, such as NLP applications? Alternatively, could the proposed forged function improve the generalization ability of the model in standard training?
3.	Could the authors elaborate on the selection process of parameter $c^r$ and explore a broader range of $c^r$ values across additional datasets?
4.	The method is efficient for certain datasets, but for large datasets like the full ImageNet, it still be computationally expensive. Do the authors have any future plans to further reduce computational requirements?

---

> ### Author Response · Authors · 2024-11-15
> **Rebuttal by Authors**
>
> **[Q3-1]** The experimental results, along with the authors' own analysis, indicate that the proposed method may reduce the classification accuracy on clean samples in some cases. While the authors discuss possible reasons, the paper could further analyze this trade-off and explore strategies to mitigate accuracy loss.
>
> **[A3-1]** We believe that we have addressed this concern in Section A.1(hyper-parameter selection): *Since most models are represented in 16 bit format, and the widths of fraction bit for FP16 format defined by IEEE-754 standard and BFloat are 10 and 7 bits, respectively, truncated errors might easily occur when performing addition on two numbers with a magnitude difference of $2^8$ or higher. On the other hand, when $c^r$ is set to $2^{-5}$, all models experience a significant drop in standard accuracy, and there is meaningless in evaluating robustness at this configuration. We suggest that the candidates of $c^r$ are $2^{-8}$, $2^{-7}$ and $2^{-6}$.*
>
> **[Q3-2]** Adjusting this parameter for new tasks could be time-consuming.
>
> **[A3-2]** As mentioned in A3-1, there are only three possible candidates for selecting $c^r$. **Scanning the entire CIFAR-10 or CIFAR-100 dataset or partial training set on ImageNet typically takes less than 5 minutes on a V100 GPU.** We can efficiently assess white-box performance using partial data from the training set, which significantly reduces the computational time. **In practice, the total time for hyperparameter search is about 1-3 hours, depending on the model size and the dataset used.**
>
> **[Q3-3]** Could the authors elaborate on the selection process of parameter and explore a broader range of $c^r$ values across additional datasets?
>
> **[A3-3]** We select the maximum value for $c^r$ as the parameter fed into the forged function. Including additional datasets could increase the risk of introducing outliers, which might undermine the effectiveness of the proposed method. Therefore, we believe that adding more datasets is unnecessary in the current implementation. Nevertheless, further investigation into other datasets could be explored in future work.
>
> **[Q3-4]** The method is efficient for certain datasets, but for large datasets like the full ImageNet, it still be computationally expensive. Do the authors have any future plans to further reduce computational requirements?
>
> **[A3-4]** We refer the reviewer to our previous responses (A3-1 and A3-2) and Section 4.2.2 (Performance Analysis on the ImageNet Dataset), where we present our evaluation of the white-box performance on ImageNet.
>
> **[Q3-5]** In the evaluation using $acc_{AA}$, are the adversarial examples generated using the original neural network architecture, or using the architecture with the proposed forged function?
>
> **[A3-5]** The adversarial examples are generated using the modified model, which incorporates the proposed forged function. For comparison, the robust accuracy against adversarial examples generated by the original models is presented in Table 7.
>
> **[Q3-6]** Can this method be extended to other tasks, such as NLP applications?
>
> **[A3-6]** The proposed method can be applied to any model that includes linear layers. However, adversarial robustness is rooted from studying robustness to adversarial examples in image domains. There are many baselines such as RobustBench, our method shows best performance. We want to make sure and focus on understanding and addressing the initial problem, before extending to other settings.
>
> **[Q3-7]** Could the proposed forged function improve the generalization ability of the model in standard training?
>
> **[A3-7]** While Lemma 2 provides a valuable property for reducing the Lipschitz constant, the current implementation is not designed to extend to standard trained models. The primary reason is that the internal features in standard trained models exhibit higher variance. Our method involves only one parameter in the forged function, and adjusting the value of $c^{th}$ can affect the results. A larger $c^{th}$ might reduce natural accuracy, while a smaller $c^{th}$ might only maintain robust accuracy, which is initially zero for standard trained models. However, there may be other forms of functions that could leverage Lemma 2 for standard trained models. Exploring such possibilities could be a valuable avenue for future research.

---

> > ### Comment · Reviewer_exHp · 2024-12-02
> >
> > Thanks for the author's reply and all my concerns are solved. I will consider improving my score.

---

### Official Review · Reviewer_TjWF · 2024-11-03

**Soundness:** 2
**Presentation:** 2
**Contribution:** 2
**Rating:** 5
**Confidence:** 4

**Summary:**

This paper proposes a robustness enhancement method based on Lipschitz continuity, aiming to improve the security of deep neural networks (DNNs) under adversarial attacks. The authors introduce a forged function that constrains the input domain of the model to reduce its Lipschitz constant, potentially enhancing robustness. This method optimizes inference without retraining the model, thus reducing computational costs. Experiments on CIFAR10, CIFAR100, and ImageNet datasets tested various model architectures like WRN and Swin Transformer, evaluating performance under both white-box and black-box attacks.

**Strengths:**

The proposed forged function based on Lipschitz continuity is implemented during the inference phase, eliminating the need for retraining or model parameter adjustments. Compared to traditional adversarial training, this method offers a significant computational cost advantage, making it more efficient for applications with limited resources.

**Weaknesses:**

1. Please distinguish between the use of `\cite{}` and `\citep{}`.
2. The authors are encouraged to open-source their code.
3. In the "Related Work" section, the authors should mention the names of methods alongside author names to aid reader comprehension.
4. In Algorithm 1, Forged Function, please add a description of the hyperparameter $c^r$ in the `require` section.
5. AutoAttack is not the latest attack method; the authors are encouraged to use more advanced black-box attack methods, as referenced in [1].
6. Although the paper emphasizes low computational overhead during the inference phase, it is recommended that the authors provide specific experimental data or quantitative analysis to compare computational overhead.

**Reference**:

[1] https://github.com/Trustworthy-AI-Group/TransferAttack

**Questions:**

1. How effective is the proposed method in more complex tasks and models, such as object detection and multimodal tasks?
2. Although the authors validated against gradient obfuscation through AutoAttack, the forged function suppresses inputs below a threshold to zero, potentially introducing gradient obfuscation under untested attack methods. Are there further theoretical or experimental supports to ensure that this method does not introduce gradient obfuscation under various attack strategies?
3. How generalizable is this method? Is it applicable to non-vision tasks?

---

> ### Author Response · Authors · 2024-11-15
> **Rebuttal by Authors**
>
> **[Q2-1]** Editorial issues such as distinguishing between the use of \cite{} and \citep{}, the authors should mention the names of methods alongside author names to aid reader comprehension in related work, and a description of the hyperparameter  c^r in the require section in Algorithm 1.
>
> **[A2-1]** We appreciate the reviewer’s suggestions and will revise the manuscript accordingly.
>
> **[Q2-2]** AutoAttack is not the latest attack method; the authors are encouraged to use more advanced black-box attack methods, as referenced in Trustworthy-AI-Group/TransferAttack.
>
> **[A2-2]** **We would like to clarify that AutoAttack is a widely accepted benchmark for evaluating white-box robustness. We believe that the comparisons presented in the paper using AutoAttack offer a fair and comprehensive evaluation of the model's performance.**
>
> The Trustworthy-AI-Group/TransferAttack, referenced by the reviewer, includes a variety of attack methods. Additionally, Table 7 presents the experimental results for transfer attacks. We would appreciate further clarification on which specific attacks from this group the reviewer recommends we include, as there are many methods, and we want to ensure we address the most relevant ones for this study.
>
> **[Q2-3]** Although the paper emphasizes low computational overhead during the inference phase, it is recommended that the authors provide specific experimental data or quantitative analysis to compare computational overhead.
>
> **[A2-3]** For the CIFAR-10 and CIFAR-100 datasets, the total training time for AWP on a V100 GPU is approximately 20+ hours. In contrast, the top three competing methods [r2-1, r2-2, r2-3] use larger models and incorporate additional data, leading to a total training time of over 200 hours. For adversarial training on ImageNet, the time required can exceed 5 days. In comparison, scanning the entire CIFAR-10 or CIFAR-100 dataset, or performing partial training on ImageNet, takes less than 5 minutes on a V100 GPU. This demonstrates that our approach incurs significantly lower computational overhead.
>
> **[Q2-4]** How effective is the proposed method in more complex tasks and models, such as object detection and multimodal tasks? How generalizable is this method? Is it applicable to non-vision tasks?
>
> **[A2-4]** The proposed method can be applied to any model that includes linear layers. However, it is important to note that adversarial robustness is rooted from studying robustness to adversarial examples in image domains. Much of the existing work and benchmarks, such as RobustBench, are specifically focused on image classification tasks. Our method has demonstrated strong performance within this domain, and at this stage, we are concentrating on addressing the core challenges related to adversarial robustness in image tasks. We believe that fully understanding and refining the approach in this well-established setting is a crucial step before extending it to more complex tasks, such as object detection, multimodal tasks, or non-vision applications. These extensions would require further investigation and adaptation, which could be valuable directions for future research.
>
> **[Q2-5]** Although the authors validated against gradient obfuscation through AutoAttack, the forged function suppresses inputs below a threshold to zero, potentially introducing gradient obfuscation under untested attack methods. Are there further theoretical or experimental supports to ensure that this method does not introduce gradient obfuscation under various attack strategies?
>
> **[A2-5]** We appreciate the reviewer’s concern regarding gradient obfuscation. However, this question can be seen as similar to asking whether future, more advanced attacks could potentially defeat current defense strategies. While we acknowledge that both attack and defense methods will continue to evolve, we believe our approach is highly effective against existing attack methods. Currently, there is no evidence suggesting that our method introduces gradient obfuscation that would render it vulnerable to untested attacks. As such, we are confident in the robustness of our method against current attack techniques.
>
> References:
>
> [r2-1] Wang, Z., Pang, T., Du, C., Lin, M., Liu, W., & Yan, S. (2023, July). Better diffusion models further improve adversarial training. In International Conference on Machine Learning (pp. 36246-36263). PMLR.
>
> [r2-2] Peng, S., Xu, W., Cornelius, C., Hull, M., Li, K., Duggal, R., ... & Chau, D. H. (2023). Robust principles: Architectural design principles for adversarially robust cnns. arXiv preprint arXiv:2308.16258.
>
> [r2-3] Bai, Y., Zhou, M., Patel, V. M., & Sojoudi, S. (2024). MixedNUTS: Training-Free Accuracy-Robustness Balance via Nonlinearly Mixed Classifiers. arXiv preprint arXiv:2402.02263

---

### Official Review · Reviewer_y1Db · 2024-11-04

**Soundness:** 3
**Presentation:** 3
**Contribution:** 3
**Rating:** 5
**Confidence:** 4

**Summary:**

This work starts from Lipschitz continuity and enhances adversarial robustness by minimizing the empirical Lipschitz constant. Specifically, the authors propose a plug-in forged function that can be inserted before each convolutional layer and MLP layer to remap the input domain of each layer into a constrained set. The theoretical guarantee of this paper primarily demonstrates that the largest singular value of the parameter matrix can serve as a loose bound for the Lipschitz constant. Experimental results show that the forged layers can bring some improvement to existing adversarial training methods.

**Strengths:**

The logic flow of this work is coherent, and the writing is clear and easy to understand. The idea of optimizing the Lipschitz constant to achieve certified adversarial robustness is also a classic approach. The implementation is very straightforward, and the proposed forged function can serve as a plug-in module that integrates with any CNN or transformer-based model.

**Weaknesses:**

Theoretical side: The bound is somewhat too loose. This paper derives the final optimization objective through the Gershgorin Circle Theorem, but this bound lacks guarantees due to the multiple assumptions made.

Empirical side: On one hand, from Tables 1 and 2, it seems that the forged function does not show significant improvements; in fact, when combined with other robust methods, the accuracy under AutoAttack even decreases. On the other hand, the robustness of the forged function itself lacks experimental support, such as how much robustness it can provide without any robustness designs (including adversarial training). An important contribution of this paper is that it incurs lower costs compared to adversarial training, so a comparison of performance with adversarial training is necessary.

Meanwhile, I suggest conducting adaptive attack against forged function to test the white-box robustness.

**Questions:**

Please see weakness part.

---

> ### Author Response · Authors · 2024-11-15
> **Rebuttal by Authors**
>
> **[Q1-1]** The bound is somewhat too loose. This bound lack guarantees due to the multiple assumptions made.
>
> **[A1-1]** We respectfully disagree with the reviewer’s comment. The only assumption we make is $A = W^\dagger W$ in Eq (10), which is introduced to simplify the notation for the subsequent proof. Additionally, we conducted an experiment to verify the behavior of the eigenvalues after applying the proposed method. The experimental results show that the average eigenvalue after applying the method is approximately 95% of the original eigenvalue. Based on this, we believe that the proposed bound is reasonable and supported by the experimental evidence. We will add this comparison in the revised version.
>
> **[Q1-2]** Tables 1 and 2, it seems that the forged function does not show significant improvements; in fact, when combined with other robust methods, the accuracy under AutoAttack even decreases.
>
> **[A1-2]** We respectfully disagree with the reviewer’s assessment. The experimental results presented in Tables 1 and 2 demonstrate that the forged function, **when integrated with existing robust methods, leads to performance improvements and achieves top-1 results on Robustbench. Furthermore, Table 4 shows that our approach outperforms several existing methods, providing stronger evidence of its effectiveness.** We believe that the results indicate the benefits of our proposed approach when combined with other techniques.
>
> **[Q1-3]** An important contribution of this paper is that it incurs lower costs compared to adversarial training, so a comparison of performance with adversarial training is necessary.
>
> **[A1-3]** We would like to emphasize that the primary contribution of this work is to improve the robustness of adversarially trained models, particularly in cases where the cost of adversarial training on large-scale datasets is prohibitive for many users. As mentioned in Section 4.2.3 (Combination with Various Models), “*by combining LefEAT model with our approach, a robust accuracy of 59.55% can be achieved, which is comparable to RST-AWP (60.04%). However, RST-AWP introduces more images from another dataset, resulting in a higher cost in each epoch. Similarly, for the CIFAR100 dataset, DefEAT with our proposed method achieves a robust accuracy of 32.11%, which is better than EffAug (31.85%), which involves more complex data augmentation during the training stage.*” We believe these results demonstrate that our approach provides comparable robustness at a significantly lower cost.
>
> **[Q1-4]** I suggest conducting adaptive attack against forged function to test the white-box robustness.
>
> **[A1-4]** We appreciate the reviewer’s suggestion. However, we would like to point out that AutoAttack, which we use in our experiments, is a widely accepted benchmark for evaluating adversarial robustness. It already includes the adaptive PGD attack. We believe that the comparisons presented in the paper using AutoAttack provide a fair and comprehensive evaluation of the model's performance.

---

### Author Response · Authors · 2024-11-22
**Kindly Reminder: Review and Discussion of Manuscript Revision**

Dear reviewers,

We would like to kindly remind you that we have submitted point-by-point responses to each of the comments provided in your review, and the revised manuscript has been uploaded for your consideration. We are truly grateful for the thoughtful feedback and constructive suggestions you have shared with us.

As the deadline for the review phase has now passed, we sincerely hope that all reviewers can participate in the discussion during the rebuttal phase, if possible. We greatly appreciate your time and effort in reviewing our manuscript.

Best regards,

Authors

---

### Meta-Review · Area_Chair_RVZ9 · 2024-12-18

**Metareview:**

The paper proposes a method to enhance adversarial robustness in deep neural networks by reducing the empirical Lipschitz constant through a novel forged function applied during inference. While the work aims to address a critical issue in AI safety and robustness, it falls short of meeting the standard for acceptance due to several significant weaknesses. The theoretical foundation is undermined by overly loose bounds derived under multiple assumptions, which limits the generalizability and rigor of the approach. Empirically, the method demonstrates marginal improvements in robustness, with results that are inconsistent across different datasets and adversarial scenarios. The robustness gains, often within 0.5%-2%, are accompanied by trade-offs in clean accuracy, which are neither systematically analyzed nor mitigated. Additionally, the novelty of the approach is questionable, as it builds upon well-established techniques such as ReLU-like thresholding without substantial innovation.

The reviewers have highlighted critical gaps, including insufficient motivation, incomplete experimental support for robustness claims, and unresolved concerns about potential gradient masking under adversarial attacks. For example, while the authors argue that their method is computationally efficient, they do not convincingly compare its performance against state-of-the-art adversarial training methods in terms of robustness-to-cost ratio. Furthermore, the paper lacks a thorough investigation of its applicability beyond vision tasks, which limits its impact and generalizability. The claims of cost-effectiveness, while appealing, are undercut by the method's limited scalability to larger datasets and complex tasks. Moreover, the rebuttal failed to adequately address concerns about empirical validation and theoretical soundness, leaving doubts about the reliability and significance of the results.

The paper does present a coherent structure and a clear narrative, with some potential for practical application due to its plug-and-play nature. However, these strengths are overshadowed by the method's incremental contribution and lack of robustness against adaptive attacks. In light of these limitations, the submission does not meet the bar for acceptance at ICLR, where originality, rigor, and impact are paramount. The work would benefit from a stronger theoretical underpinning, comprehensive empirical evaluations, and a clearer articulation of its contribution relative to existing literature.

**Additional Comments On Reviewer Discussion:**

The rebuttal period focused on several key points raised by the reviewers, with varying degrees of resolution. Reviewer y1Db pointed out concerns about the theoretical looseness of the Lipschitz constant bound and its lack of guarantees, along with limited empirical improvements and the absence of comparisons to adversarial training in terms of robustness-to-cost trade-offs. The authors responded by arguing that their theoretical assumptions were reasonable and providing additional experiments on eigenvalue behavior, but these responses did not sufficiently address the fundamental concerns about robustness validation and the method’s standalone effectiveness.

Reviewer TjWF critiqued the limited novelty of the method, highlighting its similarities to shifted ReLU functions, and requested additional evaluations against more advanced attack methods and computational overhead benchmarks. The authors partially addressed these issues by reiterating the method’s efficiency advantages during inference and justifying their use of AutoAttack. However, the lack of inclusion of more recent attack methods and insufficient experimental comparisons left these points unresolved.

Reviewer exHp raised concerns about the trade-off between robustness and clean accuracy, emphasizing the importance of hyperparameter selection and the method’s scalability to larger datasets. The authors argued that the hyperparameter search process was computationally efficient and provided detailed pruning statistics but failed to offer a deeper exploration of the accuracy trade-offs. Additionally, while scalability concerns were acknowledged, they were deferred to future work, leaving gaps in the immediate applicability of the method.

Reviewer q4ED questioned the potential for gradient masking, the empirical reductions in Lipschitz constants, and the overall novelty of the proposed approach. While the authors defended the robustness of their method against current attack techniques and provided additional eigenvalue analysis, they did not convincingly eliminate the possibility of gradient masking or establish the method’s novelty beyond incremental improvements.

Reviewer Erqp critiqued the lack of differentiation from existing Lipschitz-based defenses, the limited novelty of the forged function, and the potential increase in the Lipschitz constant with larger models. The authors attempted to clarify their method’s uniqueness by emphasizing its inference-time application and flexibility, but these arguments did not sufficiently address the fundamental issues of novelty and scalability.

In weighing these points, the main concerns—insufficient empirical robustness, limited theoretical guarantees, and incremental novelty—remained inadequately resolved. While the authors provided detailed responses, their rebuttal did not fundamentally strengthen the paper’s contributions or address critical weaknesses in a convincing manner. Consequently, these unresolved concerns significantly influenced the decision to reject the submission.

---

### Decision · Program_Chairs · 2025-01-22

Reject